# HPV and RAD51 as Prognostic Factors for Survival in Inoperable Oral and Oropharyngeal Cancer in Patients Unfit for Chemotherapy Treated with Hyperfractionated Radiotherapy

**DOI:** 10.3390/medicina59020361

**Published:** 2023-02-14

**Authors:** Zuzana Zděblová Čermáková, Pavel Hurník, David Konvalinka, Jan Štembírek, Tereza Paračková, Kamila Resová, Jakub Cvek, Tomáš Blažek, Lukáš Knybel, Martin Formánek, Mariam Gachechiladze, Markus Joerger, Alex Soltermann, Jozef Škarda, Oldřich Motyka, Jana Janoutová

**Affiliations:** 1Department of Oncology, University Hospital Ostrava, 708 52 Ostrava-Poruba, Czech Republic; 2Faculty of Medicine, University of Ostrava, 703 00 Ostrava, Czech Republic; 3Department of Clinical and Molecular Pathology and Medical Gentics, University Hospital Ostrava, 708 52 Ostrava-Poruba, Czech Republic; 4Faculty of Medicine, Masaryk University Brno, 625 00 Brno, Czech Republic; 5EUC Laboratory CGB, Inc., 703 00 Ostrava, Czech Republic; 6Department of Oral and Maxillofacial Surgery, University Hospital Ostrava, 708 52 Ostrava-Poruba, Czech Republic; 7Faculty of Medicine and Dentistry, Palacky University Olomouc, 779 00 Olomouc, Czech Republic; 8Third Faculty of Medicine, Charles University Prague, 100 00 Prague, Czech Republic; 9Department of Otorhinolaryngology and Head and Neck Surgery, University Hospital Ostrava, 708 52 Ostrava-Poruba, Czech Republic; 10Department of Medical Hematology and Oncology, Cantonal Hospital St. Gallen, 9007 St. Gallen, Switzerland; 11Facharzt Foederatio Medicorum Helveticorum (FMH) Pathologie, Pathologie Längasse, 3063 Ittingen, Switzerland; 12Faculty of Mining and Geology, VŠB-Technical University of Ostrava, 708 33 Ostrava-Poruba, Czech Republic; 13Nanotechnology Centre, VŠB-Technical University of Ostrava, Nanotechnology Centre, 708 33 Ostrava-Poruba, Czech Republic

**Keywords:** oropharyngeal cancer, oral cavity cancer, HPV, RAD51, radiotherapy

## Abstract

Introduction: The incidence of advanced oral cavity and oropharyngeal cancers is generally high. Treatment outcomes for patients, especially those unfit for comprehensive cancer treatment, are unsatisfactory. Therefore, the search for factors to predict response to treatment and increase overall survival is underway. Objective: This study aimed to analyze the presence of 32 HPV genotypes in tumor samples of 34 patients and the effect of HPV status and RAD51 on overall survival. Method: Tumor samples of 34 patients with locally advanced oropharyngeal or oral cavity cancer treated with accelerated radiotherapy in monotherapy were analyzed using reverse hybridization and immunohistochemistry for the presence of HPV and RAD51. Its effect on overall survival was examined. Results: Only two types of HPV were identified—HPV 16 (dominant) and HPV 66 (two samples). The HPV positivity was associated with a borderline insignificant improvement in 2-year (*p* = 0.083), 5-year (*p* = 0.159), and overall survival (*p* = 0.083). Similarly, the RAD51 overexpression was associated with borderline insignificant improvement in 2-year (*p* = 0.083) and 5-year (*p* = 0.159) survival. Conclusion: We found no statistically significant differences but detected trends toward improvement in the survival of HPV-positive and RAD51 overexpressing patients unfit for surgical treatment or chemotherapy treated with hyperfractionated radiotherapy. The trends, however, indicate that in a larger group of patients, the effects of these two parameters would likely be statistically significant.

## 1. Introduction

The incidence of OCOCs (oral cavity and oropharyngeal carcinomas) has increased over the last 30 years, especially among young people. This is to a large degree due to the human papillomavirus (HPV) infection, which is recognized by the World Health Organization (WHO) as a strong etiological factor for the incidence of spinocellular carcinoma in the oral cavity and oropharynx [1]. Although spinocellular carcinoma is by far the most common OCOC, other types of carcinomas, such as lymphoepithelial carcinoma or even rare sarcoma are also found in this region and must be considered during differential diagnosis [2]. Today, HPV-positive and HPV-negative tumors are considered distinct epidemiological, pathological, and clinical units. Currently, over 200 types of HPV are described. The most common classification of HPVs is according to their oncogenic potential into high-risk, probable high-risk, and low-risk HPVs. Overall, 15 known high-risk types (HR): 16, 18, 31, 33, 35, 39, 45, 51, 52, 56, 58, 59, 68, 73, and 82, 3 probable high-risk types (pHR): 26, 53, and 66, and 12 low-risk types (LR): 6, 11, 40, 42, 43, 44, 54, 61, 70, 72, and 81 have been classified. In total, 90% of HPV-positive head and neck cancers are type 16 [3], but HPV types 18 and 33 have also been related to OCOCs. In the Czech Republic, the incidence of oral and oropharyngeal cancer in 2020 was 23, and the mortality was 9.3 per 100,000 population [4].

Patients in good clinical condition with advanced disease are referred for multimodal treatment, a combination of local treatment (surgery or radiotherapy), and systemic treatment (chemotherapy, targeted cancer therapy). The number of studies with patients who are unfit for this complex treatment and are treated with radiotherapy in monotherapy is limited. We conducted a study focusing on such patients to assess their overall survival and also evaluated the dependence of survival on the presence of HPV and RAD51—a protein, the expression of which has been shown to affect the risk of solid tumor development. In addition, if the tumor is already present, it has been associated with sensitivity to oncological treatment and, in effect, with progression-free survival. RAD51 is a positive regulator of proliferation.

This study aimed to analyze the presence of 32 HPV genotypes in tumor samples from 34 patients with OCOC using reverse hybridization, to analyze the impact of the HPV status on long-term survival, and to investigate the RAD51 expression in relation to both the overall survival (OS) and progression-free survival (PFS) in our patient cohort.

## 2. Materials and Methods

### 2.1. Patients

This retrospective study included samples from 34 patients who were treated at the Department of Oncology of the University Hospital Ostrava (a center providing complex oncological care for a region covering over 1 mil. population) from 2013 to 2018. All patients signed informed consent for molecular testing of HPV genotypes. The study was approved by the appropriate Institutional Ethics Committee (IEC) of the University Hospital in Ostrava in concordance with the laws and policies of governing authorities. The group of patients comprised 24 (71%) men and 10 (29%) women 49 to 80 years old, with a median age of 62 years. In all these patients, a diagnostic biopsy was performed to allow a definitive histopathological diagnosis; these samples were subsequently used also for the additional analyses described below. General inclusion criteria were as follows: advanced, non-metastatic squamous cell carcinoma of the oropharynx and/or oral cavity and a Karnofsky index Ki > 60% with comorbidities (severe cardiovascular or renal disease or their combination) preventing the possibility of concurrent chemotherapy or targeted therapy (cetuximab). Initial staging of the disease followed recommendations and guidelines valid at the time and was based on a CT of the head and neck, an X-ray scan of the lungs, and an abdominal ultrasound. In selected cases, a whole-body PET/CT or bone scan and CT of the chest and abdomen were performed. In all patients, radical radiotherapy was indicated as a single treatment modality.

### 2.2. Radiotherapy

The patients were treated according to the HARTCIB (hyperfractionated accelerated radiotherapy with concomitant integrated boost) protocol. Target volumes included gross tumor volume of the primary tumor (GTVT) and of the metastatic lymph nodes (GTVLNs). The clinical target volume (CTV) was calculated as the sum of the GTVT + GTVLNs with a 4 mm wide isometric safety margin. Elective lymph node irradiation included bilateral lymph node levels I–V for oral cavity tumors and levels II–V for oropharyngeal tumors (clinical target volume of elective lymph nodes, CTVLNs). A 3 mm wide uniform isometric margin was used for planning target volume (PTV). Primary tumors with bulky lymph nodes and high-risk lymph node levels were irradiated with 70–75 Gy in 50 fractions, i.e., 1.4–1.5 Gy/fraction twice a day at least 6 h apart. Elective nodal levels were irradiated with a dose of 55 Gy in 50 fractions, i.e., 1.4–1.5 Gy/fraction twice a day at least 6 h apart. The spinal cord, parotid glands, pharynx, and larynx were defined as organs at risk (OAR).

### 2.3. Molecular Genetics (HPV)

Reverse hybridization of samples from 34 patients was performed to detect and identify the HPV genotype based on specific sequences in the L1 region of the HPV genome. Samples were genotyped for high-risk HPVs (16, 18, 26, 31, 33, 35, 39, 45, 51, 52, 53, 56, 58, 59, 66, 68, 70, 73, and 82) and low-risk HPVs (6, 11, 40, 42, 43, 44, 54, 61, 62, 67, 81, 83, and 89). Only two of these genotypes (HPV16 and HPV66) were detected. The formalin-fixed paraffin-embedded tissue samples were cut into 4 μm sections using a microtome. To minimize the risk of cross-contamination, the microtome knife was decontaminated between samples. Four tumor sections from each block were used for DNA isolation after xylene deparaffinization. DNA isolation was performed using the Cobas^®^ DNA Sample Preparation Kit (Roche Diagnostics GmbH, Mannheim, Germany). The concentration and purity of the isolated DNA were determined using spectrophotometry and ranged between 16 and 154 ng/μL. PCR amplification was performed using the INNO-LiPA HPV Genotyping v2 Amp amplification kit (Fujirebio, Gent, Belgium). A positive HPV6 control was included in every examination. The PCR products were subsequently used for reverse hybridization with a nitrocellulose membrane strip using the INNO-LiPA HPV Genotyping Extra II hybridization kit (Fujirebio). The results were evaluated using a reading card according to the manufacturer’s instructions.

### 2.4. Immunohistochemistry (RAD51)

For immunohistochemistry analyses, we used archived biopsy specimens that were fixed with 10% formalin and embedded in paraffin. Four-micrometer-thick sections were deparaffinized with xylene and rehydrated in a series of ethanol solutions. Heat-induced epitope retrieval was performed in 0.1 M NaOH-citrate buffer (pH 7.0) in an autoclave at 121 °C for 15 min. Endogenous peroxidase was blocked at room temperature using a 3% hydrogen peroxide in methanol for 30 min. After blocking with normal goat serum, the slides were incubated with mouse monoclonal antibody against RAD51 (MS-988-P, NeoMarkers, Fremont, CA, USA) using a 1:100 dilution of the primary antibody at 4 °C overnight. After washing, the sections were treated for 60 min at room temperature with goat–anti-mouse immunoglobulin. Staining for RAD51 was subsequently completed using the streptavidin–biotin–peroxidase complex method with diaminobenzidine as a chromogen, and the slides were counterstained with hematoxylin. Positive staining was defined as a minimum of 20% of the cancer cell nuclei showing positive nuclear staining of varying intensity (Figure 1).

### 2.5. Statistical Analysis

All statistical analyses and visualizations were performed in the R environment [5], namely using the packages FactoMineR [6], factoextra [7], survival [8], and survminer [9]. Factor analysis for mixed data was used to assess the general trends in the data as well as their relationships. Kaplan–Meier curves were constructed to illustrate the overall survival and the 2-year and 5-year survival according to both the HPV status and RAD51 overexpression (patients with 20% or greater overexpression of RAD51 will be hereinafter referred to as RAD51+ and those without RAD51 overexpression as RAD51−), the differences in the curves were assessed using the Peto and Peto modification of the Gehan–Wilcoxon test.

## 3. Results

The clinical stage of the disease was classified according to the TNM classification version 7 as follows: 5 (15%) patients were in stage III, and 29 (85%) patients were in stage IV. The median gross tumor volume (GTVT) calculated from the CT examination was 39 mL (range 5–250 mL), the median volume of the involved lymph nodes (GTVLNs) was 1.5 mL (range 0–50 mL), and the median overall tumor volume was 50 mL (range 5–250 mL). The median Karnofsky index (Ki) at the time of diagnosis was 80% (range 60–90%). The analysis showed that 12 (35%) patients had HPV-positive OCOCs, and 22 (65%) patients had HPV-negative OCOCs. Only 2 out of the 32 tested genotypes were detected in HPV-positive samples: the presence of HPV16 was detected in 9 (26%) patients and HPV66 in 2 (6%) patients. HPV typing was unsuccessful in one (3%) patient.

The median OS was 16 months (range 4–95 months, Figure 1 and Figure 2). The 2-year OS was recorded in 12 (35%) patients, of which 7 were HPV-positive (58% of HPV-positive patients) and 5 were HPV-negative (23% of HPV-negative patients). The 5-year OS was recorded in nine patients (26.4%), including four HPV-positive patients (33% of HPV-positive patients) and five HPV-negative patients (23% of HPV-negative patients). The Kaplan–Meier curve of the overall survival over 90 months depending on the HPV status of the individual patients is presented in Figure 1; the curves are visibly distinct, which shows a trend for better survival in HPV-positive patients, although their difference was not found to be statistically significant (*p* = 0.083). The same applies to shorter-term Kaplan–Meier curves of the 2-year (*p* = 0.083) and 5-year survival (*p* = 0.159) presented in Figure 2 and Figure 3, respectively.

The expression of RAD51 was immunohistochemically examined in different parts of the tumor. RAD51 overexpression was detected in 25 (74%) patients; 2 (6%) patients showed no RAD51 expression and in 7 (20%) patients, the RAD51 expression was not detectable due to the sample properties. The K–M curves depending on the RAD51 expression for the 2-year and 5-year survivals are presented in Figure 4 and Figure 5, respectively. The differences were not significant, although a trend toward better survival in patients with RAD51 overexpression was observed; this may be due to the low number of participants (coincidentally, the same *p*-values as for HPV status, i.e., *p* = 0.083 and *p* = 0.159, respectively). The median age, stage of the disease, Karnofsky index, and gross volumes in HPV positive- vs. negative- and RAD51+ vs. RAD51− patients are described in Table 1.

The factor analysis for mixed data identified the two first principal components accounting for 37% of the variance in the data. The variables GTVT+GTVLNs (gross tumor volume of tumor and gross tumor volume of metastatic lymph nodes) and GTVT were significantly positively correlated with the first principal component (20.3%), while the correlation with the overall survival (OS_months) was negative. Qualitative variables associated with this principal component included local relapse, 5-year survival status, 2-year survival status, and metastasis. Mitosis was positively correlated with the second principal component (17.1%). Qualitative variables associated with this principal component included tumor grading (G), metastasis, HPV, and 2-year survival status. The relationships between the variables, their particular categories in the case of qualitative variables, and the principal components are visualized in Figure 6 (HPV status of the individuals distinguished by color) and Figure 7 (RAD51 expression distinguished by color). In both cases, the individuals form rather distinct groups, differing mainly in their scores on the second axis (Dim 2).

## 4. Discussion

Previously published studies reported excellent results in patients treated with surgery or (chemo)radiotherapy for patients in good physical condition and with early-stage oropharyngeal cancer [10]. Markers capable of predicting treatment response and disease prognosis are still being sought. In our study, we tested the presence of HPV infection and RAD51 expression in the oral cavity and oropharyngeal cancer as predictive and prognostic markers for 2-year, 5-year, and overall survival in patients who were not eligible for surgery or chemoradiotherapy and, thus, in whom curative radiotherapy remained the only treatment option.

### 4.1. HPV Subtypes and Tumor Development

Patients with HPV-positive OCOCs are typically young people with higher socioeconomic status who are in good physical condition [11]. This might be one of the reasons why HPV-positive OCOCs are associated with better overall survival and longer progression-free survival compared to HPV-negative ones [12]. The former also show a better therapeutic response to radiotherapy (both as the sole treatment and in combination with concomitant chemotherapy) [13]. Due to the small number of positive tumors for HPV 66 in our study, it is not possible to determine whether this is a co-infection or a pathogen with oncogenic potential. Van Monsjou et al. investigated the presence of different types of HPV in oropharyngeal tumors. Of 45 samples, they found the presence of HPV66 in 1, but it did not show evidence of p16 overexpression, so it was probably not a factor in carcinogenesis [14].

### 4.2. HPV Status and Survival

No significant difference in 2-year, 5-year, or overall survival was detected between patients based on the HPV status, although there was an obvious trend for better survival in HPV-positive patients (*p* = 0.083, *p* = 0.159, and *p* = 0.083 for 2-year, 5-year, and overall survival, respectively). Garden et al. [15] evaluated survival in patients diagnosed at stages III-IVB. Their 5-year OS was 78%, which is higher than ours, but the tumors we evaluated were more advanced, the patients had additional comorbidities and were unfit for chemotherapy. Similarly, Kian Ang et al. [16], who reported 3-year survival of 64–70% for radiotherapy of patients diagnosed with stages III-IV OCOC, were able to use concomitant chemotherapy, i.e., their patients were in a generally better condition on diagnosis. The 3-year survival of HPV-positive patients in their study was better than that of the HPV-negative ones (survival: 82.4% vs. 57.1%, respectively; *p* < 0.001). This was corroborated by the results of Rischin et al. [17] who evaluated stage III-IV head and neck spinocellular carcinomas and found that the presence of p16 was associated with better 2-year survival (91% vs. 74%, *p* = 0.004).

### 4.3. RAD51 Overexpression and Survival

The DNA repair protein RAD51 is a known positive regulator of proliferation [18]. This protein is involved in nucleoprotein strand formation in single-stranded DNA, thereby mediating both homologous pairing and strand exchange reactions between single-stranded and double-stranded DNA during the repair of damaged DNA. The level of RAD51 expression has been shown to influence the risk of solid tumor development and, if a tumor is already present, to influence its sensitivity to anticancer therapy (and thus progression-free survival). RAD51 has been intensively studied as a marker of DNA repair capacity as well as chemo- and radioresistance. The literature data on RAD51 as a possible surrogate marker for radiosensitivity in head and neck cancer are, to our knowledge, not available. The result for RAD51 overexpression was similar to that of HPV positivity, especially in terms of 2- and 5-year survival—2-year and 5-year survival was borderline insignificantly improved in RAD51+ patients (*p* = 0.083 and *p* = 0.159, respectively).

Gachehciladze et al. found that immunohistochemical loss of nuclear RAD51 staining is an independent positive prognostic factor in resected non-small cell lung cancer after neoadjuvant chemo- or radiotherapy [19]. Overexpression of RAD51 was previously shown to reverse cisplatin-induced DNA damage and chemosensitivity in esophageal cancer [20], which means that the success of chemotherapy was reduced in RAD51+ patients.

Only a few papers studied RAD51 expression and its relation to treatment success in head and neck cancer. Zhao et al. reported a reduced risk of head and neck cancer associated with the G172T RAD51 polymorphism [21]. Connell et al. [22] reported that patients treated with neoadjuvant chemotherapy followed by chemoradiotherapy who had high levels of RAD51 protein had worse 2-year survival specific to head and neck cancer (33.3% versus 88.9%, respectively; *p* = 0.025).

In our study including only patients unfit for chemotherapy treatment, patients with positive RAD51 treated solely with accelerated hyperfractionated radiotherapy had slightly (though insignificantly) better 2-year survival than those with negative RAD51 (see Figure 4). This could suggest that the overexpression of RAD51 (positive regulator of proliferation) is a positive predictor of overall survival when treated with accelerated radiotherapy. Accelerated hyperfractionated radiotherapy was previously shown to be a more effective treatment option for head and neck cancer than normofractionated radiotherapy. In the case of radioresistance, dose escalation led to overcoming this problem [23].

Although our results might at the first glance be in contrast to those reported by Connell, a closer look reveals that this is not true. We must consider that these studies are to a large degree incomparable—not only because of highly different groups of patients and the different treatment regimens (chemotherapy and chemoradiotherapy were used in Connell’s study) but also due to the different classifications of RAD51 positivity. While the samples in Connell’s study were examined using immunohistochemistry and divided according to the PCI (positivity cell staining index) into 3 groups of low, moderate, and high nuclear staining, overexpression in our analysis was classified only into two groups—samples with RAD51+ (overexpression of 20% or more) and RAD51- samples (i.e., less than 20% overexpression).

The small patient population is the most obvious limitation of this study; with a larger sample, we would probably be able to demonstrate significance. However, this is largely due to the very specific group of patients evaluated in our study—it must be considered that the 34 cases were collected over a five-year period in one of the largest complex oncology centers in the Czech Republic covering a region of approx. 1 mil. population. The retrospective design of the study may be also considered a limitation, but it is fairly common in studies of rarer conditions. Lastly, due to the use of formalin-fixed paraffin-embedded samples, we were unable to determine RAD51 in all samples due to the lack of material.

## 5. Conclusions

We studied patients with advanced OCOCs who were unfit for multimodal therapy due to their overall comorbidities. We analyzed a broad battery of HPV and RAD51 and investigated whether this had an impact on the survival and success of accelerated hyperfractionated radiotherapy. We found no significant differences but detected trends toward better survival in HPV-positive patients and those with RAD51 overexpression. The fact that we have not acquired significant results is likely due to the low number of patients; the detected trends suggest that in a larger patient group, statistically significant differences would likely be observed. Hence, this study lays the basis for further investigation—a prospective study in which RAD51 expression will be correlated with potential predictors of radiosensitivity or radioresistance associated with *ataxia*-telangiectasia-mutated (ATM) and *ataxia*-telangiectasia-mutated and Rad3-related (ATR) pathways, and of overall survival.

## Data Availability

The study data will be available upon request to the corresponding author (email: pavel.hurnik@fno.cz).

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
