# Peer review of "HPV and RAD51 as Prognostic Factors for Survival in Inoperable Oral and Oropharyngeal Cancer in Patients Unfit for Chemotherapy Treated with Hyperfractionated Radiotherapy"

_medicina, 2023, doi:10.3390/medicina59020361_

Round 1

Reviewer 1 Report

Dear Authors,

 The incidence of carcinoma at the level of the oropharynx greatly increased in the last decade in comparison with other types of head and neck carcinomas through the increase in the number of cases HPV positive.

It is very important to underline the prognosis factors influencing the therapeutic choice of the multimodal treatment plan (surgery, chemotherapy, radiotherapy, and immunotherapy).

The authors focused on carcinomas HPV positive and RAD51 as prognosis factors for survival in cases with oropharynx carcinoma without surgical indication.

However, there are some aspects that need improvement in the manuscript:

1.      Please change the title adding the words “for survival in” before the words inoperable oral.

2.      Line 44 – instead of the term “high” try to approximate the incidence in the Czech Republic.

3.      Line 46 – add the word “increase” before the word overall, for better clarity of the message.

4.      Line 50 – please correct the typo “zanalyzed”.

5.      Line 60 – please correct the explanation between the brackets with “oral cavity and oropharyngeal carcinomas”.

6.      Line 67 – after low-risk instead of a column use a full stop and continue with another sentence enumerating the high-risk types of HPV.

7.      Line 70 – after reference number 2 use a full stop and continue with another sentence about types 18 and 33.

8.      Line 76 – please explain the abbreviation RAD51.

9.      Line 78 – please delete the word “thus” for clarity.

10.   Line 94 – please use the term biopsy instead of “excision”.

11.   Line 98 – the most frequent abbreviation is Ki instead of “KI”

12.   Line 113 and 114 – please refer to the N staging in the TNM staging instead of “bulky lymph nodes and high-risk lymph nodes”.

13.   Line 146 – use RAD51 instead of Rad51.

14.   Line 154 – use instead of HNSCC the abbreviation OCOC.

15.   Line 166 – insert after the words version 7 the particle “as follows”.

16.   Line 230 – reference the affirmation regarding the combined regimen for early-stage cancer.

17.   For the Discussion section also mention the differential diagnosis of oropharyngeal carcinomas, because there is a great variety of cell lines that determine these carcinomas. One possible reference article could be Vrînceanu D, Dumitru M, Ştefan AA, Mogoantă CA, Sajin M. Giant pleomorphic sarcoma of the tongue base - a cured clinical case report and literature review. Rom J Morphol Embryol. 2020 Oct-Dec;61(4):1323-1327. doi: 10.47162/RJME.61.4.34. PMID: 34171081; PMCID: PMC8343483.

18.   Line 245 – add “for HPV66” before in our study.

19.   Line 274 – add a blank space between the full stop and RAD51.

20.   Line 278 – instead of “a” use and for better clarity.

21.   Line 297 – Please underline the fact that the expression of RAD51 is a negative predictive factor for the overall survival if the patients are treated only with chemotherapy and in the cases overexpressing it is necessary to increase the regimen of radiation therapy.

22.   Please explain the abbreviation FFPE.

23.   Please explain the abbreviations ATM, ATR, OS.

24.   Please format the bibliography according to MDPI style.

It is known the role of a positive prognostic factor of HPV infection in OCOC is due to the fact that these tumors develop in young patients, and the tonsillar infection with HPV induces a higher immune response in the patient. The manuscript discusses the role of HPV and RAD51 in OCOC advanced cases with an impact on the choice of treatment and survival.

Reviewer 2 Report

The subject of the study is very interesting, though the manuscript semana incomplete. There are sections to be filled, I.e. “abstract”. References are not standardized for any style. Regarding the content of the manuscript, despite the absence of significance, the correlation the studied markers with cancer prognosis is interesting and can be better explored for authors. The “Conclusion” section can be improved with solid statements.

Reviewer 3 Report

The sample size is too small to draw any conclusions. In addition, the adequacy of statistical analysis methods should be reassessed. For example, (line 155) "a trend was observed" , the description is confusing as well as the factor analysis for mixing data in line 163.

Round 2

Reviewer 2 Report

Authors reviewed the manuscript correcting and/or improving the text. Manuscript is now complete and interesting results have been presented.